# The protective effect and immunomodulatory ability of orally administrated *Lacticaseibacillus rhamnosus* GG against *Mycoplasma pneumoniae* infection in BALB/c mice

Huanbing Long[☯], Guiting He[☯], Jiarong He, Ting feng Du, Pengxiao Feng, Cuiming Zhu[ID]*

Hunan Provincial Key Laboratory for Special Pathogens Prevention and Control, Institute of Pathogenic Biology, Hengyang Medical College, University of South China, Hengyang, Hunan, People's Republic of China

☯ These authors contributed equally to this work.
* nhzhucuiming@usc.edu.cn

**Data Availability Statement:** All relevant data are within the paper.

## Abstract

*Mycoplasma pneumoniae* represents one of the significant etiologies of community-acquired pneumonia in pediatric patients. However, clinical treatment of *M. pneumoniae* infection in children has encountered challenges due to the escalating resistance to quinolones. Numerous studies have highlighted the potential of probiotic lactobacillus administration in boosting immune responses to bacterial and viral respiratory infections. In this study, the protective efficacy of pre-oral administration of *Lacticaseibacillus rhamnosus* GG (LGG), *Limosilactobacillus reuteri* F275, *Lactiplantibacillus plantarum* NCIMB 8826, *L. plantarum* S1 or *L. plantarum* S2 was evaluated in the BALB/c mice model; it was observed that among these five strains of lactobacillus, the supplementation of LGG exhibited the most significant protective effect against *M. pneumoniae* infection. Moreover, when administered orally, both live LGG and heat-inactivated LGG have demonstrated efficacy in reducing the burden of *M. pneumoniae* in the lungs and alleviating pulmonary inflammation. Oral supplementation with LGG resulted in the inhibition of neutrophil recruitment into the lungs and increased recruitment of alveolar macrophages in *M. pneumoniae*-infected mice. Additionally, LGG supplementation led to increased production of IL-10 and secretory IgA (sIgA), while suppressing the levels of IL-1β, IL-6, IL-17A, and TNF-α in the lungs of mice infected with *M. pneumoniae*. The data suggests that supplementation with LGG can modulate immune responses, decrease pathogen load, and alleviate inflammatory injury in the lungs of *M. pneumoniae*-infected mice.

## Introduction

*Mycoplasma pneumoniae* is an important microorganism associated with community-acquired pneumonia (CAP), accounting for 10%-30% of CAP cases during periods of endemicity among children [1, 2]. However, during epidemics, the infection rate can escalate

**Funding:** This study was supported by the National Natural Science Foundation of China (No. 31970177) and the Research Project of Hunan Health Commission (D202310008129). The funders had no role in study design, data collection and analysis, decision to publish, or preparation of the manuscript.

**Competing interests:** The authors have declared that no competing interests exist.

to 50% or higher in individuals with respiratory tract infections. *M. pneumoniae* pneumonia (MPP) typically follows a self-limiting and spontaneously resolvable course. Nonetheless, a considerable number of clinical instances of MPP progress to severe, refractory, and potentially life-threatening pneumonia [3, 4]. Moreover, *M. pneumoniae* is an essential trigger for pediatric wheezing and worsening of asthma [5, 6]. Owing to the absence of cell walls, *M. pneumoniae* is not susceptible to β-lactam antibiotics; the recommended drugs for treating MPP mainly encompass macrolides, tetracyclines, and fluoroquinolones [7]. Since tetracyclines and fluoroquinolones have obvious side effects in infants and children, macrolides are preferred for treating MPP in children. However, macrolide resistance is prevalent in many countries and varies by geographical region [8, 9]. Additionally, the lack of effective vaccines [10], has made the clinical management of MPP in children extremely challenging. Therefore, searching for alternative/additional treatments and new prevention strategies is essential for preventing MPP.

Lactobacillus species, which are gram-positive anaerobic bacteria, are commonly found in the human gastrointestinal tract, oral cavity, vagina, and respiratory tract [11]. Certain strains of Lactobacillus serve as widely accepted probiotics in various commercial products and as essential fermenting agents in dairy products, playing a significant role as natural immunobiotics, expressing robust immunomodulatory capacity and antipathogenic activity [12]. Increasing evidence supports that certain Lactobacillus strains exhibit potential in alleviating bacterial and viral respiratory tract infections. In clinical trials, for instance, oral administration of *L. reuteri* SD 2112 [13], *L. plantarum* L-137 [14, 15], *L. plantarum* DR7 [14, 15], *Lacticaseibacillus casei* Shirota [16, 17], or *L. casei* DN-114001 [16, 17] in infants or adults can reduce the incidence of upper respiratory infections and alleviate the clinical symptoms. Moreover, oral administration of mixed probiotics, specifically *Lactobacillus* KABP022/KABP023/KABP033, has been shown to reduce the risk of respiratory failure in patients with novel coronavirus pneumonia by eight times [18, 19], and Bifidobacterium lactobacillus triplex live tablet (containing *Bifidobacterium Longum*, *L. Bulgaricus*, and *Streptococcus thermophiles*) oral treatment shortened the duration of diarrhea, reduced time to achieving a negative nucleic acid test and the inflammation indexes including procalcitonin and C-reactive protein in patients with severe novel coronavirus pneumonia [20]. In preclinical animal studies, it has been observed that the oral administration of *L. rhamnosus* CRL1505 [21], *L. rhamnosus* M21 [22], and *Lactobacillus johnsonii* [21] elicits an immune response against the respiratory syncytial virus, influenza virus, or pneumococcal infection, resulting in enhanced pathogen clearance and reduced lung tissue injury. Additionally, oral administration of *L.casei* CRL431 [23] and *L. plantarum* CIRM653 [24] has shown effectiveness in decreasing the occurrence and duration of pathogen infections, as well as in reducing pathogen loads associated with *Pseudomonas aeruginosa*, *Streptococcus pneumoniae*, and *Klebsiella pneumoniae*. Furthermore, LGG attenuates lung injury and inflammatory response in experimental sepsis [25]. In previous research, we discovered that *L. casei* CNRZ1874, a novel strain of *L. casei* [26, 27] and a potential probiotic from the collection of the China General Microbiological Culture Collection Centre (CGMCC), can alleviate MPP by activating M1 alveolar macrophages [28]. *L. casei* is a common probiotic genus on the qualified presumption of the safety (QPS) list in Europe and China [29]. However, it is well known, that the probiotic effects are strain-specific, further assessment of the safety of using *L. casei* CNRZ1874 is necessary. Employing immunobiological agents to prevent or treat respiratory infections may present an attractive alternative strategy. The use of probiotic lactobacillus to prevent or treat *M. pneumoniae* infections may be an ideal alternative strategy.

Therefore, in the present study, we conducted an assessment of the protective effects of orally administered Lactobacillus strains, including LGG, *L. reuteri* F275, *L. plantarum*

NCIMB 8826, *L. plantarum* S1, and *L. plantarum* S2, against *M. pneumoniae* infection in BALB/c mouse models. Among these five Lactobacillus strains, LGG demonstrated the most robust protective effects. Additionally, we conducted a preliminary investigation to assess the potential adjuvant effect of orally administering live and heat-inactivated LGG in the prevention of *M. pneumoniae* infection, with a specific focus on the immunoregulatory impact of LGG on *M. pneumoniae*.

## Materials and methods

### Main reagents

Pleuropneumonia-like organism (PPLO) broth, FITC anti-mouse CD45 Antibody, PE anti-mouse Ly-6G Antibody, PerCP-Cy5.5 anti-mouse CD11c Antibody, A647(APC) anti-mouse Siglec-F Antibody, Rat anti-mouse Fc receptor (CD16/CD32) Antibody (clone 2.4G2), PE-Cy7 anti-mouse CD25 Antibody, Alexa Fluor (APC) anti-mouse Foxp3 Antibody and Intracellular Fixation & Permeabilization Buffer Set Kit (BD Biosciences, USA); de Man, Rogosa and Sharpe (MRS) medium and LACATE Dehydrogenase(LDH) Activity Assay Kit (Solarbio, Beijing, China); TRIzol reagent (Invitrogen, Carlsbad, CA, USA); FastKing RT Kit (With gDNase) (TIANGEN, China); ChamQ Universal SYBR qPCR Master Mix (Vazyme, China); BCA Protein Assay Kit (Epizyme, China); albumin assay kit (Nanjing Jiancheng Biology, China); TNF-α, IL-1β, IL-6, IL-10 and IL-17A ELISA kit (Thermo Fisher, USA), TGF-β and MPO ELISA kit (Bioswamp, China); Horseradish peroxidase (HRP)-conjugated Goat Anti-mouse IgM and IgA (Proteintech, USA); PMA, ionomycin and brefeldin A (Sigma–Aldrich, USA); PE anti-mouse CD4 Antibody, FITC anti-mouse CD8a Antibody, APC anti-mouse IFN-γ Antibody, PerCPCy5.5 anti-mouse IL-4 Antibody (BioLegend, USA).

### Mice

Five to six-week-old specific pathogen-free (SPF) mice male BALB/c mice (Gempharmatech Co., Ltd.) were housed in an SPF barrier environment at the Animal Management Office of the University of South China. Before the experiment, mice were housed for five days to acclimate to the environment. The experimental protocols were sanctioned by the Animal Experimentation Ethics Committee of the University of South China (ethics code: USC202205XS38), and all procedures rigorously adhered to the ethical guidelines set forth by the World Organization for Animal Health.

After being orally administered, the mice were anesthetized prior to infection and sample collection steps with Isoflurane, and the anesthetized mice were intranasally inoculated with *M. pneumoniae*. The mice were euthanized for experimental analysis by cervical dislocation after anaesthesia. The pain and discomfort caused by the experiment on the animals, as well as whether there is any abnormal behaviour in the animals, are observed closely. Whenever feasible, mice were euthanized by $CO_2$ asphyxiation as soon as they displayed signs of severe infection.

### Lactobacilli administration

LGG (ATCC 53103), *L. plantarum* S1, and *L. plantarum* S2 were kindly provided by Professor Xiaohua Chen of Hengyang Normal University, China. Both strains of *L. reuteri* F275 (DSM20016, ATCC 23272, or CIP109823) and *L. plantarum* NCIMB 8826 (ATCC BAA-793) were purchased from Shanghai Beinuo Biological Co., LTD. Five strains of Lactobacillus were propagated in MRS broth at 37°C for 12–16 h. Subsequently, the organisms were harvested, washed with sterile phosphate-buffered saline (PBS), and reconstituted in PBS to maintain a

bacterial concentration of $5 \times 10^9$ colony-forming units (CFU)/ml. The heat inactivation of LGG was assessed at 70˚C for 1 h using a temperature-controlled water bath [30]. The mice were orally administered $10^9$ CFU of live LGG (n = 25), heat-inactivated LGG (n = 20), *L. reuteri* F275 (n = 5), *L. plantarum* NCIMB 8826 (n = 5), *L. plantarum* S1 (n = 5), and *L. plantarum* S2 (n = 5) in 200 μl of PBS daily for seven days, while the control group (n = 25) received 200 μl of PBS via gavage.

## *M. pneumoniae* preparations and inoculations

The *M. pneumoniae* M129 strain (ATCC 29342) was maintained at the Institute of Pathogen Biology, Hengyang Medical College, University of South China, China. The organism was revitalized in 5 ml of PPLO broth medium at 37˚C for 48 hours, then 2 ml of the bacterial solution was transferred to T-150 cell culture flasks containing 20 ml of PPLO broth medium and subcultured at 37˚C until the broth acquired an orange hue, which took approximately 120 hours. The upper cell culture media in the culture flasks was discarded, and 4 ml of fresh PPLO broth medium was added. The bottom of the vial was gently scraped with a cell scraper, and the collected material was centrifuged at 10,000×g for 15 minutes. After being washed once with sterile PBS, the organism was suspended in sterile PBS to a concentration of $2.5 \times 10^9$ CFU/ml. Following administration, all *M. pneumoniae*-infected mice were anesthetized and intranasally inoculated with $10^8$ CFU of *M. pneumoniae* in 40 μl of PBS the subsequent day. On the 3rd and 7th days post-infection (3 dpi and 7 dpi), the mice were anesthetized, and organ dissection was meticulously performed to further examine the immunoprotective effect and immunomodulatory mechanism of LGG to *M. pneumoniae* infection.

## The burden of *M. pneumoniae*

The bronchoalveolar lavage (BAL) fluid was harvested and then centrifuged at 900×g for 10 min to separate the cells and the supernatant [21]. Subsequently, 50 μl of fresh BAL fluid supernatant was evenly spread on the PPLO agarose medium and placed in a constant temperature incubator containing 95% $N_2$ and 5% $CO_2$ at 37˚C for 7–10 days. Following incubation, the bacterial colonies were counted under an inverted microscope.

Total RNA was extracted from lung homogenates using TRIzol reagent, followed by reverse transcription with a FastKing RT kit to obtain cDNA. Subsequently, real-time PCR was employed to detect the *M. pneumoniae p*1 adhesin gene. The *p*1 gene primers used in this experiment were forward 5 ’-CGCCCAAAAGATGAATGAC-3’, reverse 5’- TGTCCCCCATTAC ACGTTC-3’) [31]. The internal reference gene for this study was mouse β-actin. The quantification of the *p*1 gene was conducted using the 2-ΔΔCt method, utilizing the standard concentration of the plasmid pUC57-*p*1, which was synthesized by Sangon (Shanghai, China).

## Lung tissue injury parameters

The lower right lobe of the murine lung was excised and subsequently fixed in a 4% paraformaldehyde solution for 48 h. After fixation, the tissue was subjected to paraffin embedding, followed by sectioning and Hematoxylin and Eosin (HE) staining. The application of a double-blind protocol enabled the observation of pathological inflammatory changes within the pulmonary tissue. The assessment of histopathology scoring (HPS) involved the analysis of the inflammatory cell presence and extent of infiltration in the alveoli and bronchi, as well as the effusion in the bronchial and bronchiolar lumens, perivascular infiltration, and interstitial pneumonia [32]. An intensity score in each of three categories was determined: peribronchiolotis (0 none to 3 severe), alveolitis (0 none to 3 severe), and interstitium of the lung (0 none to

3 severe). A weighted intensity score was then determined based on the percentage of lung involvement.

The total protein, albumin content, LDH activity, and myeloperoxidase (MPO) activity in the BAL fluid were tested. The total protein concentration in the BAL fluid was determined using a bicinchoninic acid (BCA) assay. The albumin level was assessed using an albumin assay kit based on albumin binding to bromocresol green. LDH activity was analyzed using an LDH activity assay kit. The myeloperoxidase (MPO) levels in the BAL fluid were measured using an indirect enzyme-linked immunosorbent assay (ELISA).

## Flow cytometry

Pulmonary cells were obtained using established techniques [33]. Lung tissue was homogenized in 1 ml of sterile PBS using a 70-micron cell strainer and centrifuged at 1200×g for 5 minutes. The resulting pellet was resuspended in 35% Percoll and centrifuged at 700×g for 15 minutes. The pellet was then treated with red blood cell lysis buffer to eliminate erythrocytes. Following centrifugation, the cellular pellet was resuspended in PBS. The pulmonary cells and cells in the BAL fluid underwent flow cytometry analysis after staining with fluorescently labelled antibodies (anti-mouse CD45, Ly-6G, CD11c, and Siglec-F antibody). Leukocytes were identified as CD45+ cells, neutrophils as CD45+Ly6G+ cells, and alveolar macrophages as CD45+CD11c+Siglec-F+ cells [34].

To analyze the activation of Th1 and Th2 cells, pulmonary cells were incubated at 37˚C for 4 hours with 50 ng/ml phorbol myristate acetate (PMA), 1 µg/ml ionomycin, and 1 µmol/ml brefeldin A. The Fc receptors of the cells were then blocked, and the cells were stained with fluorescently labelled CD4 antibody and CD8a antibody. Subsequently, the cells were fixed with permeabilization buffer and stained with fluorescently labelled anti-mouse IFN-γ and IL-4 antibodies. To analyze the T regulatory cell (Treg) response, pulmonary cells were blocked with a rat anti-mouse Fc receptor antibody and incubated with fluorescently labelled anti-mouse CD4 antibody and CD25 antibody. Upon fixation and disruption of the membranes, the cells underwent incubation with an APC-conjugated anti-mouse Foxp3 antibody. Subsequently, all cell samples were assessed utilizing an LSRII flow cytometer (BD Bioscience), and the resultant data were analyzed using FlowJo.

## ELISA

The concentrations of TNF-α, IL-1β, IL-6, IL-10, IL-17A, and TGF-β in BAL fluid were quantified using enzyme-linked immunosorbent assay (ELISA). Absorbance readings were obtained at 450 nm, and the concentrations were determined by extrapolation from the standard curve.

The concentrations of immunoglobulin M (IgM) in the serum and IgA in the BAL fluid were determined using ELISA without any dilution of BAL fluid, while a 1:2 dilution of the serum. Specifically, 15 µg/mL of *M. pneumoniae* soluble antigens, prepared according to established protocols [35], were coated on microtiter plate wells and subsequently blocked. The binding of IgM and IgA to the coated antigens was probed using horseradish peroxidase (HRP)-conjugated goat anti-mouse IgM and IgA antibodies, respectively. Colorimetric detection was achieved by adding tetramethylbenzidine (TMB) solution, followed by the termination of the reaction with a stop solution. The absorbance measured at 450 nm was used to represent the levels of IgM and IgA.

## Statistical analyses

The statistical analyses were performed using SPSS 8.0 software. The data was reported as the mean ± standard deviation (SD). Variations were evaluated using the Bonferroni multiple comparison test with one-way ANOVA. Statistical significance was established at $P<0.05$.

# Results

## LGG shows the most substantial protective impact on mice infected with *M. pneumoniae*

The effects of intestinal supplementation with LGG, *L. reuteri* F275, *L. plantarum* S1, *L. plantarum* S2, and *L. plantarum* NCIMB 8826 on *M. pneumoniae* infection in mouse models were investigated. This involved determining the pathogen loads and evaluating lung tissue on the third day post the pathogen infection. This specific day point was chosen because it coincided with the peak of lung tissue inflammation and the highest organism loads in the infected mouse lungs. Compared to the control group, oral supplementation with LGG, *L. reuteri* F275, and *L. plantarum* S1 resulted in a significant decrease in *M. pneumoniae* loads (Fig 1A), the number of inflammatory cells and infiltration areas (Fig 1B), and HPS (Fig 1C). In contrast, oral treatment with *L. plantarum* NCIMB 8826 and *L. plantarum* S2 did not yield significant decreases in the pathogen burden. The LDH activity, total protein, and albumin indicated increased permeability and cellular damage to the bronchoalveolar capillary barrier. The results showed supplementation with LGG and *L. reuteri* F275 resulted in a significant decrease in the total protein content of BAL fluid and LDH activity in mice. However, the BAL Fluid albumin content did not exhibit a significant variance among the groups of mice (Fig 1D). These observations suggest that, in comparison to *L. plantarum* NCIMB 8826, *L. reuteri* F275, *L. plantarum* S1, and *L. plantarum* S2, intestinal supplementation of LGG confers the most substantial protective effect on *M. pneumoniae* infection in the mouse model.

## Both live LGG and heat-inactivated LGG demonstrated significant efficacy against *M. pneumoniae* infection

To investigate whether colonization of LGG to intestinal mucosal epithelial cells is essential for alleviating *M. pneumoniae* respiratory tract infections, we examined the protective effects of pre-supplementation with live LGG stain or a heat-inactivated LGG (or dead LGG) strain against *M. pneumoniae* infection. At 3 dpi, oral administration of live or dead LGG substantially reduced the *M. pneumoniae* load in the BAL fluid (Fig 2B). The *p*1 gene copies, as detected by real-time PCR, decreased in both live LGG and dead LGG pretreated mice (Fig 2B), consistent with the colony culture results. Moreover, at 3 dpi, the lungs of mice orally pretreated with live LGG or heat-killed LGG exhibited reduced numbers of inflammatory cells and infiltration area, along with more intact alveolar structures (Fig 2C), HPS was lower compared to control mice (Fig 2D), and both live LGG and heat-killed LGG treatments decreased LDH activity, total protein, and albumin levels in *M. pneumoniae* infected mice (Fig 2E). However, there was no significant difference in pathogen loads and lung jury between mice pretreated with live LGG and those treated with heat-killed LGG strain. At 7 dpi, the pathogen loads, the number of inflammatory cells, the infiltration area, and HPS were reduced in the lung tissues of mice supplemented with live or heat-inactivated strains of LGG. BAL Fluid LDH activity was reduced in mice supplemented with live LGG strains, but total protein and albumin contents were not significantly different between the groups of mice.

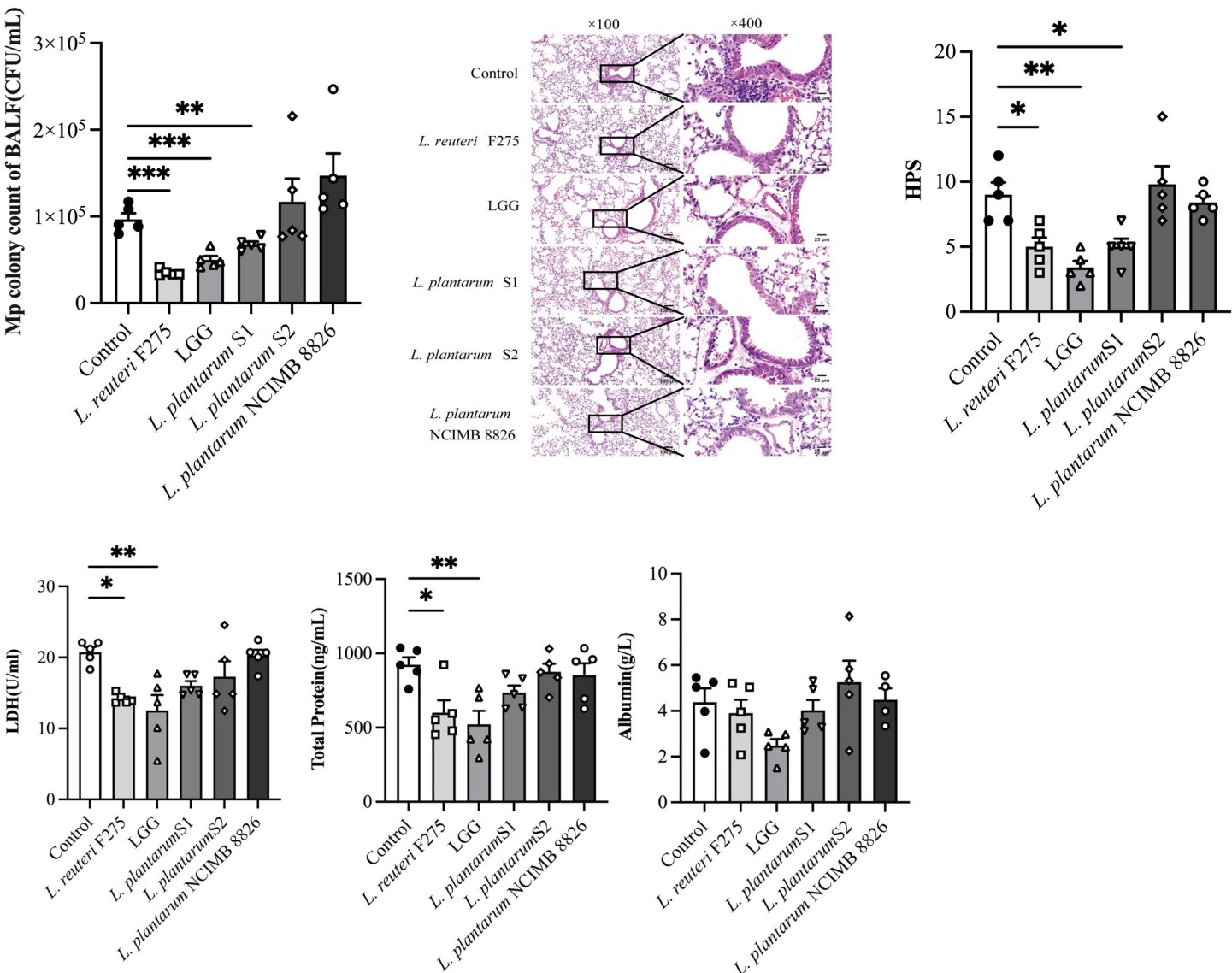

**Fig 1. Supplementation of five Lactobacillus strains affected the colonization of *M. pneumoniae* and lung injuries.** On the 3rd day after *M. pneumoniae* infection, (a) the *M. pneumoniae* colony counts in mouse BAL fluid, (b) Lung histopathology (representative images at ×100 and ×400 magnification for each experimental group, (c) HPS, (d) LDH activity, as well as the concentrations of total proteins and albumin in BAL fluid (N = 30, n = 5). Each value is indicative of the mean ± SD. * $p < 0.05$, ** $p < 0.01$, *** $p < 0.001$.

## LGG supplementation modulated the recruitment of neutrophils and alveolar macrophages

Immune cells in the BAL fluid and lungs generally indicate pathological changes in MPP. To assess the impact of LGG treatment on the recruitment of inflammatory cells into lung tissue following *M. pneumoniae* infection, the distribution of immune cells in the BAL fluid (Fig 3B) and lungs (Fig 3C) was examined at 3 dpi, corresponding to the acute phase of *M. pneumoniae* infection. Intranasal inoculation of M. pneumoniae into mice led to an increase of total leukocytes with abundant neutrophils in BAL fluid compared to uninfected mice. The supplementation of LGG effectively reduced the total leukocyte and neutrophil profiles in the BAL fluid of *M. pneumoniae*-infected mice. Additionally, the reduction in the

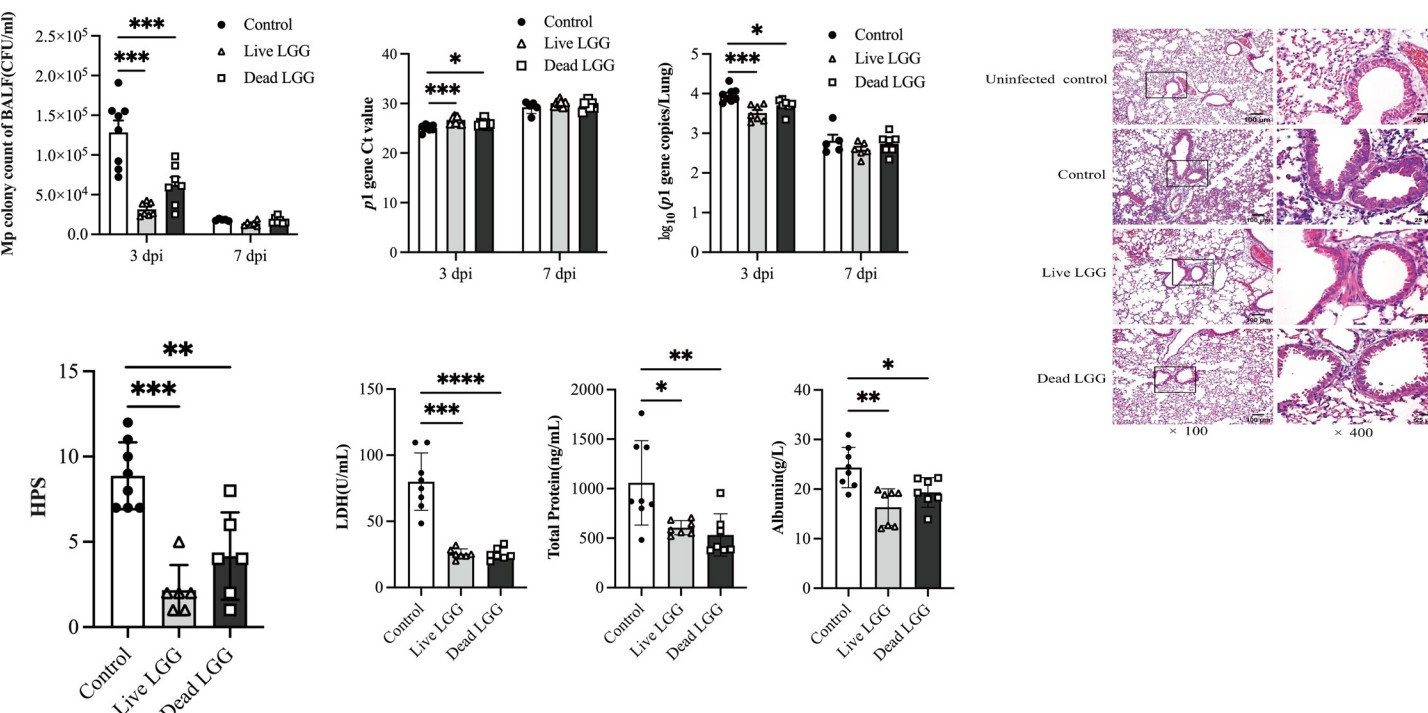

**Fig 2. Both live and heat-inactivated LGG supplementation attenuated lung tissue injury caused by *M. pneumoniae* infection.** Pre-oral supplementation with live or heat-inactivated strains of LGG at 3 dpi and 7 dpi (a) the *M. pneumoniae* colony counts in mouse BAL fluid, (b) *p*1 gene copies in lung tissue homogenate detecting by RT-PCR, (c) Lung histopathology (representative images at ×100 and ×400 magnification for each experimental group, (d)HPS, (e) LDH activity, as well as the concentrations of total proteins and albumin in BAL fluid. Each value is indicative of the mean ± SD. * *p* < 0.05, ** *p* < 0.01, *** *p* < 0.001.

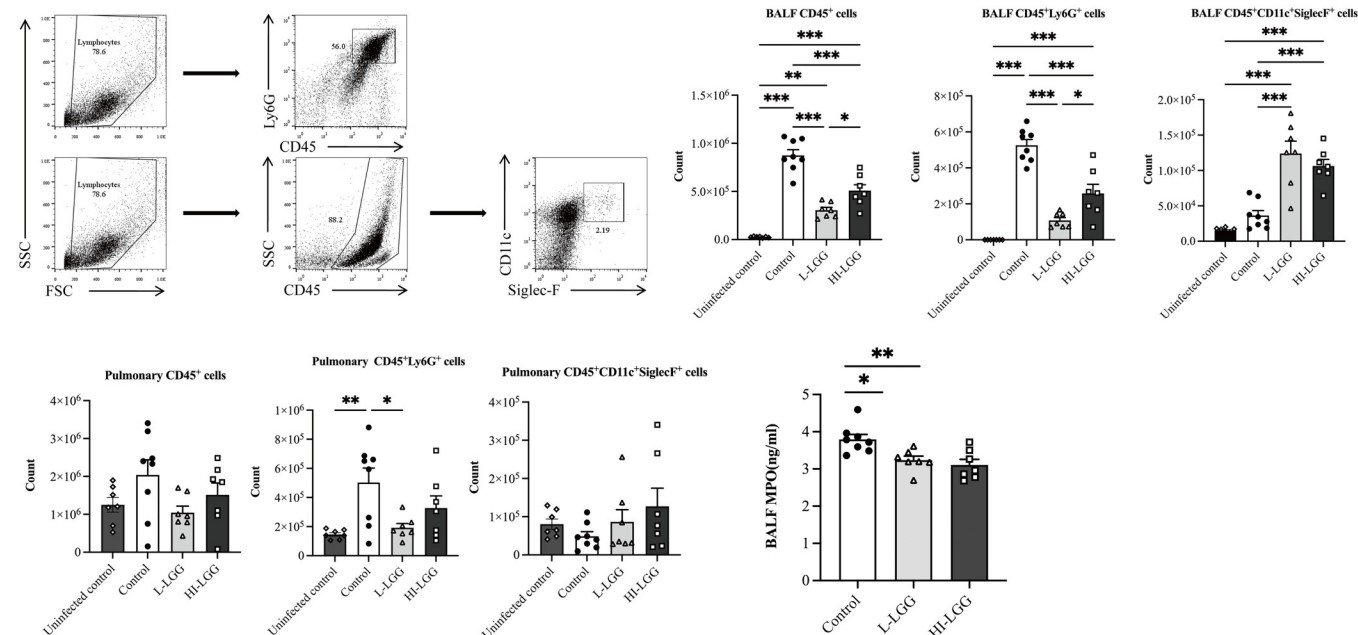

**Fig 3. Supplementation with LGG effectively influenced the recruitment of innate immune cells following *M. pneumoniae* infection.** Following oral administration of LGG (LGG), the mice were intranasally infected with *M. pneumoniae* and euthanized at 3 dpi. (a) Neutrophils (CD45+Ly6G+) and macrophages (CD45+CD11c+Siglec-F+) in BAL fluid and lung tissue were identified using flow cytometry, (b and c) Cell populations in the BAL fluid and lung were characterized, (d) Myeloperoxidase (MPO) activity in BAL fluid was quantified. Each value represents the mean ± SD. * *p* < 0.05, ** *p* < 0.01, *** *p* < 0.001.

number of neutrophils in the BAL fluid of live LGG-administrated mice was greater than that in the BAL fluid of dead LGG-treated mice. Both live and dead LGG-supplemented mice showed an increase in the number of alveolar macrophages in BAL fluid. Compared to the PBS gavage group, the number of neutrophils in the lung tissue of the live LGG treatment group was decreased. The changes in the numbers of pulmonary alveolar macrophages were similar to those in the BAL fluid, with no obvious differences observed. Furthermore, neutrophil infiltration in the BAL fluid was assessed by quantifying MPO [36]. The production of MPO in the BAL fluid of mice subjected to live or dead LGG pretreatment all exhibited a reduction, and no difference was found between mice pre-treated with live LGG and those treated with heat-killed LGG strain (Fig 3D).

### LGG supplementation regulated the secretion of cytokines

To further examine the immunoregulatory implications of orally administered LGG on the inflammatory response triggered by *M. pneumoniae*, the levels of proinflammatory cytokines such as TNF-α, IL-1β, IL-6, and IL-17A, as well as anti-inflammatory cytokines IL-10 and TGF-β in BAL fluid were analyzed at 3 dpi. Compared to PBS-pretreated mice, oral supplementation with live LGG or heat-killed LGG reduced the secretion of TNF-α, IL-1β, and IL-6. Notably, live LGG treatment was more effective in diminishing the levels of IL-6 compared to heat-killed LGG treatment. Moreover, a discernible reduction in levels of IL-17A was exclusively observed in mice administered live LGG. Both live LGG and dead LGG supplementation elevated IL-10 secretion, while no significant difference in TGF-β secretion was observed among these three groups (Fig 4).

### LGG supplementation promoted the production of IgA in the BAL fluid

To examine the humoral immunity of *M. pneumoniae*-infected mice with or without LGG supplementation, serum, and BAL fluid was obtained at 7 dpi to assess mycoplasma-specific IgM and IgA. Results demonstrated that compared to the PBS gavage group, both live and dead LGG supplementation increased the IgA concentration in the BAL fluid (Fig 5A). However, LGG treatment did not impact the production of mycoplasma-specific serum IgM (Fig 5A). Both sIgA and IgM levels were elevated in mice with intranasal with *M. pneumoniae* compared to the baseline level of uninfected mice but the IgM level of dead LGG-supplemented group.

To determine whether administering LGG affects T-cell-mediated immunity during *M. pneumoniae* infection, CD4+ and CD8+ T cells that produce IL-4 and IFN-γ, as well as CD4 + T cells expressing the specific transcription factor Foxp3 in pulmonary cells at 7 dpi were examined. The analysis of CD4+T and CD8+T cell detection in lung tissue revealed that supplementation with LGG did not produce a significant impact on the proportions of CD4+IL-4 +, CD4+IFN-γ+, and CD8+IFN-γ+ T cells (Fig 5B). Furthermore, there were no significant differences in the proportions of Foxp3+ Tregs (Fig 5C). These findings suggest that the administration of LGG is insufficient to modulate the T-cell immune response elicited by *M. pneumoniae* in BALB/c mice.

## Discussion and conclusion

LGG is a strain of Lactobacillus that was discovered in 1983 within the intestinal tract of a healthy human. This strain is widely regarded as one of the safest probiotics and has garnered recognition for its safety in numerous countries [37]. In China, LGG distinguishes itself as one of eight lactobacilli that have obtained official approval for integration into food products intended for infants and toddlers, which positively influence digestive health [38]. Moreover, it exhibits marked capabilities in immune modulation [39] and demonstrates exceptional

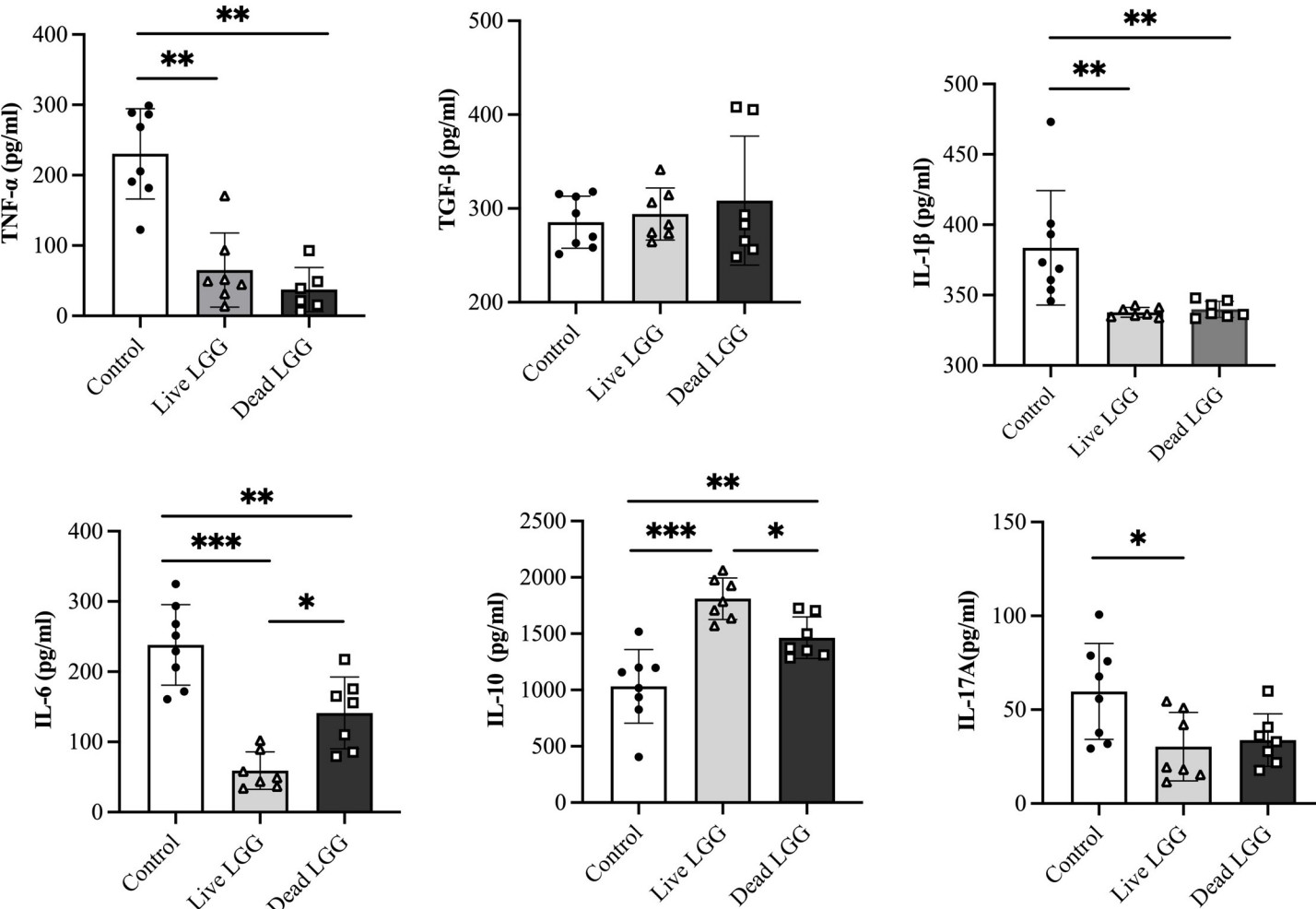

**Fig 4. Supplementation with LGG modulated the production of cytokines induced by *M. pneumoniae*.** Following oral administration of LGG, the mice were intranasally infected with *M. pneumoniae* and were subsequently sacrificed at 3 dpi. Cytokine secretion in the BAL fluid was quantified using ELISA. Each value represents the mean ± SD. * $p < 0.05$, ** $p < 0.01$, *** $p < 0.001$.

antibacterial and antiviral properties [40, 41]. Pertinent research has indicated that the oral administration of LGG serves as an efficacious strategy in reducing the incidence and duration of respiratory tract infections in children [42], as well as in mitigating pathogen loads, including those associated with H1N1 influenza virus [43], *P. aeruginosa* [44] and *Moraxella catarrhalis* [45], as observed in murine models or in vitro experiments. Our study demonstrated that during the early stages of *M. pneumoniae* infection, supplementing with LGG has been shown to alleviate pulmonary inflammation damage and reduce the pathogen loads in the lungs. It conferred the most significant protective effect on *M. pneumoniae*-infected BALB/c mice compared to *L. reuteri* F275, *L. plantarum* NCIMB 8826, *L. plantarum* S1, and *L. plantarum* S2.

According to several studies, even after LGG is no longer alive, it still exhibits immunomodulatory activity, possibly due to the immune-boosting properties of bacterial components like extracellular lipoteichoic acid and peptidoglycans [46, 47]. Arellano-García, Laura, et al confirmed that both live and heat-inactivated LGG can partially prevent hepatic oxidative stress and the inflammatory response induced by a diet that promotes liver fat accumulation in rats

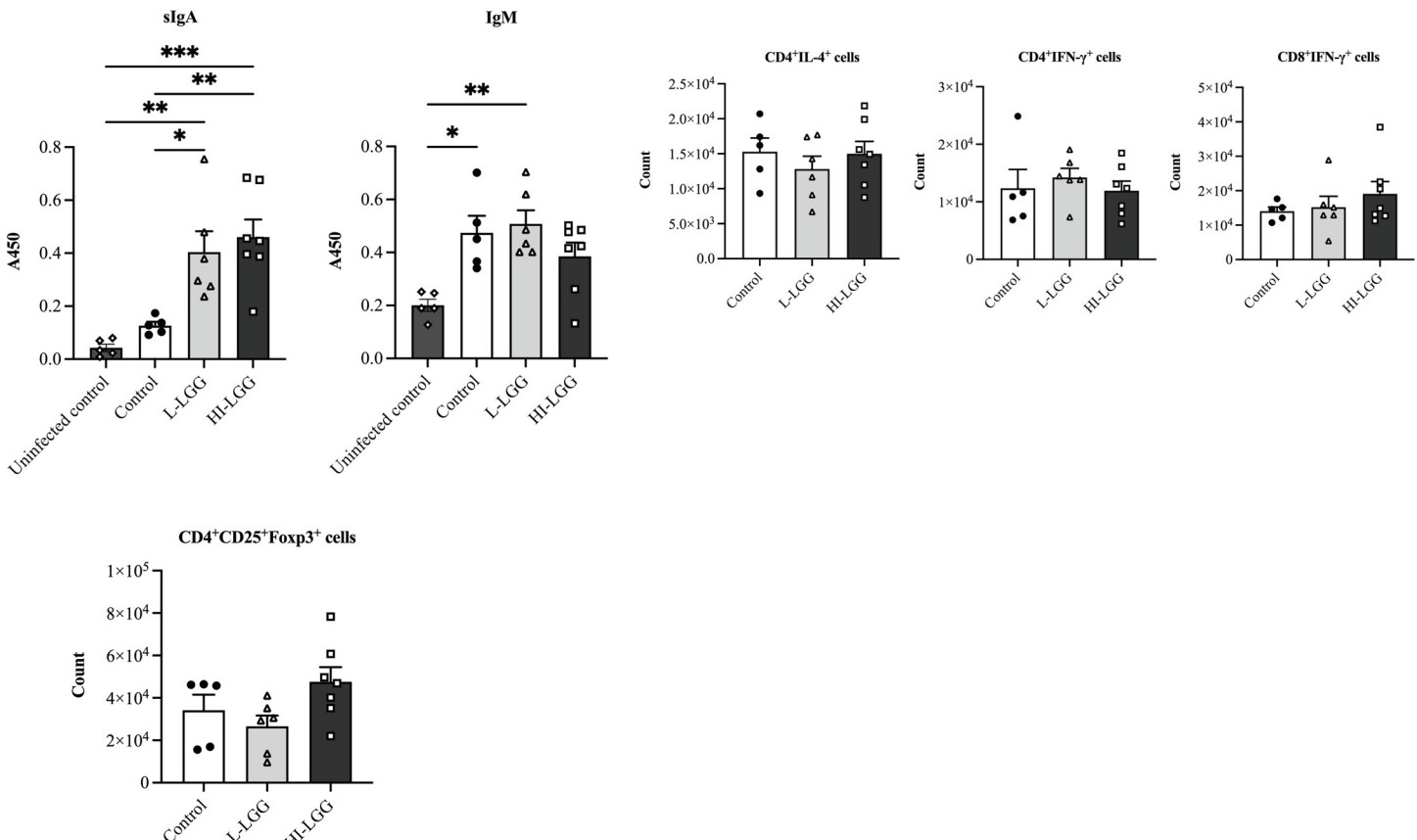

**Fig 5. The regulation of LGG supplementation of the adaptive immune triggered by *M. pneumoniae*.** Following oral administration of LGG, the mice were intranasally infected with M. *pneumoniae* and subsequently sacrificed at 7 dpi. (a) Analysis of *M. pneumoniae*-specific IgA levels in the BAL fluid and IgM levels in the serum. (b) The profiles of CD4+ and CD8+ T cells secreting IFN-γ and IL-4. (c) The profiles of CD4+ T cells expressing Foxp3. Each value represents the mean ± SD. * $p < 0.05$, ** $p < 0.01$.

[48]. Additionally, Kumpu M et al found that the reduction in the rhinovirus infection rate did not significantly differ between live and heat-inactivated LGG [49]. The findings of our study indicate that both live and heat-inactivated LGG were effective in promoting the clearance of *M. pneumoniae* and attenuating pulmonary inflammation. Due to its greater safety profile and easier storage requirements, heat-inactivated LGG may be more practical for preventing MPP.

The infection caused by M. *pneumoniae* results in the recruitment of neutrophils and alveolar macrophages. Alveolar macrophages play a principal role in the immune response against *M. pneumoniae* infection [50]. However, the available evidence indicates that neutrophils offer minimal assistance in effectively clearing *M. pneumoniae* [36–51]. Conversely, they exacerbate pulmonary damage and are linked to the onset of MPP [52], playing a crucial role in the development of refractory MPP [53]. Our study revealed that pre-treatment of mice with live or dead LGG before *M. pneumoniae* infection resulted in the suppression of neutrophil recruitment into the lungs while concurrently facilitating the accumulation of alveolar macrophages following pathogen exposure. A reduction in neutrophils played a significant role in alleviating the inflammation caused by *M. pneumoniae*, while an increase in alveolar macrophages facilitated the clearance of the pathogen burden. The literature suggests that TNF-α, IL-1β, and IL-6 are significant proinflammatory cytokines in *M. pneumoniae* infection [54]. Furthermore, IL-17A has been identified as a key mediator of lung tissue injury following *M. pneumoniae*

infection [55, 56]. Conversely, TGF-β and IL-10 are anti-inflammatory cytokines that fulfill crucial negative regulatory functions within the immune response [57, 58]. In the present study, the results demonstrate that the oral administration of live or dead LGG resulted in a significant reduction in the production of TNF-α, IL-1β, and IL-6. Specifically, the administration of live LGG led to a notable decrease in IL-17A. Additionally, an elevated level of IL-10 was observed in the BAL fluid of mice administered live or dead LGG. Live LGG demonstrated superior efficacy in suppressing neutrophil recruitment and IL-6 production while eliciting more pronounced macrophage recruitment to lung tissue and IL-10 production. These changes in cytokine secretion within the lungs align with the recruitment of leukocytes and the amelioration of pulmonary inflammation and tissue damage in the early *M. pneumoniae* infection stage.

Prior research indicates that the oral administration of LGG can modulate adaptive immunity to combat some respiratory infections [59–61]. Specifically, LGG treatment has been observed to suppress allergen-induced airway inflammation by promoting Th2-cell responses [59] and alleviate *P. aeruginosa*-induced pneumonia by eliciting Treg responses [44]. LGG can produce allergen-specific IgA in patient saliva, thus stimulating oral mucosal responses [60]. Additionally, LGG intervention has been shown to significantly increase IgG levels in children suffering from rotavirus diarrhea [61]. Our investigation observed that the supplementation of LGG did not exert regulatory effects on *M. pneumoniae*-specific T-cell immunity in BALB/c mice. Antigen-specific IgM, an antibody produced early during *M. pneumoniae* infection, is an important component in the fight against the pathogen. In this study, oral administration of LGG did not demonstrate modulation in serum IgM production following M. pneumoniae infection. This may be attributed to the inadequacy of LGG treatment to influence systemic immunity within the scope of this investigation, or it is plausible that an extended duration is required to detect mycoplasmal-specific IgM, despite typical detection occurring 7 days after *M. pneumoniae* infection. Following oral supplementation of live or dead LGG, an increase in the secretion of *M. pneumoniae*-specific IgA in BAL fluid was observed. This finding suggests that LGG has the potential to enhance mucosal immunity during mycoplasma infection by impeding the organism's adherence to respiratory epithelial cells. In our previous study, we found that oral treatment with *L. casei* CNRZ1874 did not affect the production of myco-plasma-specific antibodies in C57BL/6J mice [28]. This difference may be due to using different Lactobacillus strains and mouse strains.

In conclusion, our current study provides evidence that oral administration of LGG before *M. pneumoniae* infection demonstrates a capacity to mitigate lung burden and attenuate lung tissue damage through modulation of neutrophil and alveolar macrophage recruitment, cytokines secretion, and IgA production. Supplementation with LGG confers a strong protective effect on *M. pneumoniae*-infected BALB/c mice and may serve as an alternative therapeutic approach and a novel prevention strategy for MPP. However, it is imperative to conduct further research to explore the influence of oral LGG treatment on long-term protective effects against *M. pneumoniae* infection, as well as to examine the potential effects of immunomodulatory components on lactobacilli, such as lipoteichoic acid and peptidoglycan.

## Conclusions

Oral administration of both live and heat-inactivated LGG before *M. pneumoniae* infection can inhibit the infiltration of neutrophils and increase the recruitment of alveolar macrophages in the lungs. It can also suppress the production of IL-1β, IL-6, IL-17A, and TNF-α and elevate the secretion of IL-10 and sIgA, which contribute to the clearance of *M. pneumoniae* and the relief of MPP in the BALB/c mouse model.

## Acknowledgments

We would like to express our gratitude to the Hunan Provincial Key Laboratory for Special Pathogens Prevention and Control for providing experimental facilities, to other laboratory staff for their assistance in our experiment, to Professor Deli Xin from the Institute of Tropical Medicine at Beijing Friendship Hospital affiliated with Capital Medical University for her support in the experiment, to Professor Chen Xiaohua from Hengyang Normal University for presenting the LGG, L. plantarum S1, and S2 strains involved in this study.

## Author Contributions

**Conceptualization:** Huanbing Long, Guiting He, Cuiming Zhu.

**Formal analysis:** Huanbing Long, Guiting He, Jiarong He, Ting feng Du, Pengxiao Feng.

**Funding acquisition:** Cuiming Zhu.

**Methodology:** Huanbing Long, Guiting He, Jiarong He.

**Writing – original draft:** Guiting He.

**Writing – review & editing:** Huanbing Long, Cuiming Zhu.

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
