## [Decision Letter · Decision Letter 0]

25 Jun 2024

PONE-D-24-11070The protective effect and immunomodulatory ability of oral supplementation of Lactobacillus rhamnosus  GG against  Mycoplasma pneumoniae infection in BALB/c micePLOS ONE

Dear Dr. Cuiming,

Thank you for submitting your manuscript to PLOS ONE. After careful consideration, we feel that it has merit but does not fully meet PLOS ONE’s publication criteria as it currently stands. Therefore, we invite you to submit a revised version of the manuscript that addresses the points raised during the review process.

We look forward to receiving your revised manuscript.

Kind regards,

Palash Mandal

Academic Editor

PLOS ONE

Journal Requirements:

3. Thank you for submitting the above manuscript to PLOS ONE. During our internal evaluation of the manuscript, we found significant text overlap between your submission and previous work in the [introduction, conclusion, etc.].

Please revise the manuscript to rephrase the duplicated text, cite your sources, and provide details as to how the current manuscript advances on previous work. Please note that further consideration is dependent on the submission of a manuscript that addresses these concerns about the overlap in text with published work.

[If the overlap is with the authors’ own works: Moreover, upon submission, authors must confirm that the manuscript, or any related manuscript, is not currently under consideration or accepted elsewhere. If related work has been submitted to PLOS ONE or elsewhere, authors must include a copy with the submitted article. Reviewers will be asked to comment on the overlap between related submissions (http://journals.plos.org/plosone/s/submission-guidelines#loc-related-manuscripts).]

We will carefully review your manuscript upon resubmission and further consideration of the manuscript is dependent on the text overlap being addressed in full. Please ensure that your revision is thorough as failure to address the concerns to our satisfaction may result in your submission not being considered further.

'This study was supported by the National Natural Science Foundation of China (No. 31970177) and the Research Project of Hunan Health Commission (D202310008129).'

Please state what role the funders took in the study.  If the funders had no role, please state: 'The funders had no role in study design, data collection and analysis, decision to publish, or preparation of the manuscript.' 

Additional Editor Comments:

Dear Authors,

For your guidance, the reviewers' comments are included below.

Thank you for giving us the opportunity to consider your work.

Specific concerns expressed during peer review were:

<xxxx; p="" t="xxxx,">

Reviewers' comments:

Reviewer's Responses to Questions</xxxx;>

**Comments to the Author**

1. Is the manuscript technically sound, and do the data support the conclusions?

Reviewer #1: Yes

Reviewer #2: Yes

2. Has the statistical analysis been performed appropriately and rigorously? 

Reviewer #1: Yes

Reviewer #2: Yes

3. Have the authors made all data underlying the findings in their manuscript fully available?

Reviewer #1: Yes

Reviewer #2: Yes

4. Is the manuscript presented in an intelligible fashion and written in standard English?

Reviewer #1: Yes

Reviewer #2: Yes

5. Review Comments to the Author

Reviewer #1: The manuscript presents original research about the role of probiotics in preventing Mycoplasma pneumoniae pneumonia. In particular, the authors observed protective effects on a single oral L. rhamnosus GG dose that affected innate immunity rather than adaptive immunity.

The manuscript is well-designed and presented, the experiments are well-described, and the conclusions are accordant to the results obtained.

I have some comments to make in order to clarify and correct the manuscript.

According to the International Scientific Association for Probiotics and Prebiotics, the name "Lactobacillus rhamnosus" should be changed to "Lacticaseibacillus rhamnosus."

The animals were maintained under SPF conditions. Were they born under the same conditions? Did you use SPF mice?

For the anti-M. pneumoniae antibodies evaluation (lines 160-161), please include the dilution employed for serum and BAL (probably BAL was used undiluted, but serum must have been diluted). Moreover, signals look quite low, especially for serum IgM (of course, it depends on the dilution factor). Can you re-evaluate serum antibodies, including some samples of normal, uninfected mice? Including a line in the graph showing the "background" signal of the ELISA would be helpful in understanding if there are no differences in specific antibodies produced or if there are no specific antibodies produced.

I assume that comparisons in brackets (t=xxxx, p<xxxx; p="" t="xxxx,">The authors claim that specific molecules might be involved in the effects of probiotics since even the heat-killed bacteria present those effects. They suggest that peptidoglycan and exopolysaccharide may explain the effects. However, exopolysaccharides are molecules commonly secreted to the medium, and they cannot be presented at high concentration in heat-killed bacteria. I suggest exploring the potential role of lipoteichoic acid instead of exopolysaccharide (Nutrients Volume 14, Issue 3, 2022 Article number 723: is a research of my laboratory, the reference is not to cyte in this manuscript, but just to support my comment).

Line 91 "...(MRS) medium and LACATE dehydrogenase (LDH)..."</xxxx;>

Reviewer #2: The paper reported that oral supplementation with either live or dead LGG can protect mice against intranasal challenge with Mycoplasma pneumoniae. The protection is evidenced by the reduced M. pneumoniae lung burden, reduced lung tissue damage accompanied by reduced recruitment of neutrophils, and increased macrophages. Additionally, there is a reduction in the production of cytokines such as IL-1β, IL-6, and TNF-α, along with an elevation in IL-10 and IgA in the lungs. These results are all descriptive.

The main concern is the lack of a non-challenge control group in the experiment. The study only includes groups challenged with M. pneumoniae but does not have a non-challenge control. The absence of data from normal mice makes interpreting the results challenging.

Another concern is the inconsistency in the timing of data collection. Data presented in Figs 1-4 are at 3 dpi, while data in Figs 5-7 are at 7 dpi. This inconsistency makes it difficult to correlate the factors affecting the outcomes. It would be better to include testing for Figs 1-4 at 7 days and for Figs 5-7 at 3 days.

L50 Lacticaseibacillus should be Lactobacillus

L52. Please provide the reference related the safety of L. casei.

L104. How was the efficacy of the heat inactivation of LGG verified?

L126. The score system should be detailed.

L177. Is there any live LGG recovered or LGG gene detected in BAL from any of the three treatment groups?

Figue 5 How about the IgG levels against Mp? This data is collected at 7 dpi and might not be relevant to the positive effects observed at 3 dpi.

L312. It might need to test at a longer time point instead of 7 days.

6. PLOS authors have the option to publish the peer review history of their article (what does this mean?). If published, this will include your full peer review and any attached files.

Reviewer #1: **Yes: **Daniel GONZÁLEZ MAGLIO

Reviewer #2: No

---

## [Author Response · Author response to Decision Letter 0]

4 Aug 2024

Manuscript ID: PONE-D-24-11070 ---Revised for PLOS ONE.

Entitled " The protective effect and immunomodulatory ability of orally administrated Lacticaseibacillus rhamnosus GG against Mycoplasma pneumoniae infection in BALB/c mice” by Long H et al.

We want to thank the editor for the good comments on our manuscript. Following the comments, we have revised our manuscript. Hopefully, you will find our revised manuscript suitable for publication in PLOS ONE. Our point-to-point responses to the editor's comments are listed as follows.

Response: We have revised the manuscript according to PLOS ONE's style requirements.

Response: We have provided the information regarding the experiments involving animals.

3. Thank you for submitting the above manuscript to PLOS ONE. During our internal evaluation of the manuscript, we found significant text overlap between your submission and previous work in the [introduction, conclusion, etc.]. We would like to make you aware that copying extracts from previous publications, especially outside the methods section, word-for-word is unacceptable. In addition, the reproduction of text from published reports has implications for the copyright that may apply to the publications.

Please revise the manuscript to rephrase the duplicated text, cite your sources, and provide details as to how the current manuscript advances on previous work. Please note that further consideration is dependent on the submission of a manuscript that addresses these concerns about the overlap in text with published work. 

[If the overlap is with the authors’ own works: Moreover, upon submission, authors must confirm that the manuscript, or any related manuscript, is not currently under consideration or accepted elsewhere. If related work has been submitted to PLOS ONE or elsewhere, authors must include a copy with the submitted article. Reviewers will be asked to comment on the overlap between related submissions (http://journals.plos.org/plosone/s/submission-guidelines#loc-related-manuscripts).]

We will carefully review your manuscript upon resubmission and further consideration of the manuscript is dependent on the text overlap being addressed in full. Please ensure that your revision is thorough as failure to address the concerns to our satisfaction may result in your submission not being considered further.

Response: Thank you for pointing it out. We have modified these possible text overlaps in our resubmitted manuscript. 

Response: Thank you. We have provided the correct grant numbers in forms submitted online.

5. Please state what role the funders took in the study. If the funders had no role, please state: 'The funders had no role in study design, data collection and analysis, decision to publish, or preparation of the manuscript.'

Response: The role the funders took in the study has been stated.

Response: Thanks for the suggestion. Our ethics statement only appeared in the Methods section of our revised manuscript.

Manuscript ID: PONE-D-24-11070 ---Revised for PLOS ONE.

Entitled " The protective effect and immunomodulatory ability of orally administrated Lacticaseibacillus rhamnosus GG against Mycoplasma pneumoniae infection in BALB/c mice” by Long H et al.

We would like to thank the reviewers and the editor for the good comments on our manuscript. Following the comments, we have revised our manuscript and carefully checked and edited English grammar, spelling, and sentence structure by a professional language editor. Moreover, we supplement the experiment of “The investigation of intestinal supplementation with Five strains of lactobacillus on Mycoplasma pneumoniae infection in mouse models”. Hopefully, you will find our revised manuscript suitable for publication in PLOS ONE. Our point-to-point responses to the reviewers' comments are listed as follows.

Reviewer 1 comments：

1. According to the International Scientific Association for Probiotics and Prebiotics, the name "Lactobacillus rhamnosus" should be changed to "Lacticaseibacillus rhamnosus."

Response: Thank you for your correction. We have replaced "Lactobacillus rhamnosus GG" with "Lacticaseibacillus rhamnosus GG".

2. The animals were maintained under SPF conditions. Were they born under the same conditions? Did you use SPF mice?

Response: The BALB/c mice used in this experiment were purchased from Gempharmatech Co., Ltd, which has an SPF-level experimental animal production license and an experimental animal usage license. Mice are packed in special autoclaved cardboard with transparent viewing windows and filter membranes. Then mice were housed in the SPF Animal Laboratory at the University of South China. So, the mice used in our experiment are SPF mice. 

3. For the anti-M. pneumoniae antibodies evaluation (lines 160-161), please include the dilution employed for serum and BAL (probably BAL was used undiluted, but serum must have been diluted). Moreover, signals look quite low, especially for serum IgM (of course, it depends on the dilution factor). Can you re-evaluate serum antibodies, including some samples of normal, uninfected mice? Including a line in the graph showing the "background" signal of the ELISA would be helpful in understanding if there are no differences in specific antibodies produced or if there are no specific antibodies produced.

We express our sincere appreciation for your valuable advice. In our endeavor to detect anti-M. pneumoniae antibodies, we employed undiluted BAL fluid and a 1:2 dilution for the serum. Given the low immunogenicity of M. pneumoniae, the resultant antibody detection signals are correspondingly low [1,2]. In the results section, we included the levels of serum IgM and sIgA in uninfected mice, as well as the levels of macrophages and neutrophils in BAL fluid and lung tissue to illustrate the "background" signal.

Reference:

[1] Chu HW, et al. Repeated respiratory Mycoplasma pneumoniae infections in mice: effect of host genetic background. Microbes Infect, 2006 Jun;8(7):1764-72. doi: 10.1016/j.micinf.2006.02.014.

 [2] Wubb el L, et al. Mycoplasma pneumoniae pneumonia in a mouse model. J Infect Dis

. 1998 Nov;178(5):1526-9. doi: 10.1086/314439.

4. I assume that comparisons in brackets (t=xxxx, p) The authors claim that specific molecules might be involved in the effects of probiotics since even the heat-killed bacteria present those effects. They suggest that peptidoglycan and exopolysaccharide may explain the effects. However, exopolysaccharides are molecules commonly secreted to the medium, and they cannot be presented at high concentration in heat-killed bacteria. I suggest exploring the potential role of lipoteichoic acid instead of exopolysaccharide (Nutrients Volume 14, Issue 3, 2022 Article number 723: is a research of my laboratory, the reference is not to cite in this manuscript, but just to support my comment).

Response: We sincerely appreciate the valuable comments. We re-analyzed the causes of the heat-killed LGG present anti-M. pneumoniae effects, and replaced exopolysaccharide to lipoteichoic acid, also provide the reference. Moreover, lipoteichoic acid could be a valuable line of thought for our next experiment. 

5.Line 91 "...(MRS) medium and LACATE dehydrogenase (LDH)..."

Response: We have replaced "Lacate" with " LACATE ".

Reviewer 2 comments：

1. The main concern is the lack of a non-challenge control group in the experiment. The study only includes groups challenged with M. pneumoniae but does not have a non-challenge control. The absence of data from normal mice makes interpreting the results challenging.

Response: Thank you for pointing it out. We had a non-challenge control group in the experiment. We are sorry the results of the non-challenge control group were not arranged. We supplemented the result of the non-challenge control group in the revised manuscript. 

2. Another concern is the inconsistency in the timing of data collection. Data presented in Figs 1-4 are at 3 dpi, while data in Figs 5-7 are at 7 dpi. This inconsistency makes it difficult to correlate the factors affecting the outcomes. It would be better to include testing for Figs 1-4 at 7 days and for Figs 5-7 at 3 days.

Response: At 3 dpi, this specific day point was chosen because it coincided with the peak of lung tissue inflammation and the highest organism loads in the infected mouse lungs. At 7 dpi, to examine the humoral immunity of M. pneumoniae-infected mice with or without LGG supplementation and to determine whether administering LGG affects T-cell-mediated immunity during M. pneumoniae infection, adaptive immunity detection typically occurred 7 days after M. pneumoniae infection, or even an extended duration was needed.

3. L50 Lacticaseibacillus should be Lactobacillus.

Response: According to the International Scientific Association for Probiotics and Prebiotics, the name "Lactobacillus casei" was changed to “Lacticaseibacillus casei”. Lacticaseibacillus casei CNRZ1874 was abbreviated as L. casei CNRZ1874 in the revised manuscript. 

4. L52. Please provide the reference related the safety of L. casei.

Response: We have supplemented some information on the safety of L. casei and provided the related reference. L. casei is a common probiotic genus on the qualified presumption of the safety (QPS) list in Europe and China [1]. However, it is well known, that the probiotic effects are strain-specific, further assessment of the safety of using L. casei CNRZ1874 is necessary.

Reference:

[1] EFSA BIOHAZ Panel, Koutsoumanis, K., Allende, A., Alvarez- Ordonez, A., Bolton, D., Bover- Cid, S. et al. (2022) Updated list of QPS- recommended biological agents for safety risk assessments carried out by EFSA.

5. L104. How was the efficacy of the heat inactivation of LGG verified?

Response: After the LGG was assessed at 70°C for 1 h using a temperature-controlled water bath, the organism was cultured in MRS solid culture for 48 hours, and no colony was observed. This is according to the results of the literature (doi: 10.1016/j.ijfoodmicro.2013.04.022), which showed that after 20 min heat treatment at 60 °C, colony-forming units of LGG were significantly decreased. The temperatures 70 °C and 80 °C were already lethal for strains [1].

Reference:

[1] du Toit E, Vesterlund S, Gueimonde M, Salminen S. Assessment of the effect of stress-tolerance acquisition on some basic characteristics of specific probiotics. Int J Food Microbiol 2013, 165(1):51-56. doi: 10.1016/j.ijfoodmicro.2013.04.022.

6. L126. The score system should be detailed.

Response: Thanks for the good comment. We have detailed described the scoring system.

7. L177. Is there any live LGG recovered or LGG gene detected in BAL from any of the three treatment groups?

Response: Thanks for the good suggestion. We did not detect BAL LGG from any of the three treatment groups, this is the deficiency of our experiment. According to literature reports, the gut epithelial barrier separates the internal intestinal milieu from the luminal environment, thereby ascertaining the permeability of nutrients and other molecules as well as exerting a protective role by arresting the entry of microbes and toxic compounds [1]. Studies have reported that commensal Lactobacilli spp. maintain gut barrier integrity which depends upon the multi-protein complexes such as tight junctions, gap junctions, adherents, and desmosomes [2]. Additionally, findings manifested that it was by expanding mesenteric CD103+DCs and accumulating mucosal Tregs that LGG treatment orally contributes to protecting against OVA-induced allergic airway inflammation [3]. While the appropriate tests are still needed to ascertain whether LGG can be identified in BAL.

Reference:

[1] Rastogi S, Singh A. Gut microbiome and human health: Exploring how the probiotic genus Lactobacillus modulate immune responses. Front Pharmacol. 2022 Oct 24;13:1042189. doi: 10.3389/fphar.2022.1042189.

[2] Kocot AM, Jarocka-Cyrta E, Drabińska N. Overview of the Importance of Biotics in Gut Barrier Integrity. Int J Mol Sci. 2022 Mar 7;23(5):2896. doi: 10.3390/ijms23052896.

[3] Zhang J, Ma JY, Li QH, Su H, Sun X. Lactobacillus rhamnosus GG induced protective effect on allergic airway inflammation is associated with gut microbiota. Cell Immunol. 2018 Oct;332:77-84. doi: 10.1016/j.cellimm.2018.08.002.

8. Figue 5 How about the IgG levels against Mp? This data is collected at 7 dpi and might not be relevant to the positive effects observed at 3 dpi.

Response: IgM is typically the first antibody to be produced after M. pneumoniae infection. In the M. pneumoniae-infected BALB/c mice model, IgM peaked on day 7 after a primary infection. IgG is the major antibody class seen late after the first infection, the reaction of mycoplasma with mycoplasma-specific IgG is necessary for phagocytosis of the organism by macrophages (opsonization). Generally, a noticeable IgG response was observed at day 14 post M. pneumoniae infection in BALB/c mice[1,2], so at 7 dpi, we did not detect the IgG levels. Thanks for the good comment, in our future studies, we will further refine our experiment, and detect the most important specific IgG antibodies against mycoplasma infection on day 14 after the pathogen infection.

Reference:

[1] Chu HW, et al. Repeated respiratory Mycoplasma pneumoniae infections in mice: effect of host genetic background. Microbes Infect, 2006 Jun;8(7):1764-72. doi: 10.1016/j.micinf.2006.02.014.

 [2] Wubb el L, et al. Mycoplasma pneumoniae pneumonia in a mouse model. J Infect Dis

. 1998 Nov;178(5):1526-9. doi: 10.1086/314439.

9. L312. It might need to test at a longer time point instead of 7 days.

Response: Thank you for your good suggestion. In the M. pneumoniae-infected BALB/c mice model, IgM peaks on day 7 after a primary infection. But at this time point, IgG, the most important antibody against mycoplasma infection, is almost undetectable on 7 dpi. A longer time point instead of 7 dpi is needed. In future experiments, we will test the antibodies on day 14 dpi or a longer point.

---

## [Decision Letter · Decision Letter 1]

28 Aug 2024

The protective effect and immunomodulatory ability of orally administrated Lacticaseibacillus rhamnosus GG against Mycoplasma pneumoniae infection in BALB/c mice

PONE-D-24-11070R1

Dear Dr. Zhu Cuiming,

We’re pleased to inform you that your manuscript has been judged scientifically suitable for publication and will be formally accepted for publication once it meets all outstanding technical requirements.

Kind regards,

Palash Mandal

Academic Editor

PLOS ONE

---

## [Editor Report · Acceptance letter]

8 Oct 2024

PONE-D-24-11070R1 

PLOS ONE

Dear Dr. Cuiming, 

I'm pleased to inform you that your manuscript has been deemed suitable for publication in PLOS ONE. Congratulations! Your manuscript is now being handed over to our production team.

Kind regards, 

on behalf of

Prof. Palash Mandal 

Academic Editor

PLOS ONE